# Radiomics in Oesogastric Cancer: Staging and Prediction of Preoperative Treatment Response: A Narrative Review and the Results of Personal Experience

**DOI:** 10.3390/cancers16152664

**Published:** 2024-07-26

**Authors:** Giovanni Maria Garbarino, Michela Polici, Damiano Caruso, Andrea Laghi, Paolo Mercantini, Emanuela Pilozzi, Mark I. van Berge Henegouwen, Suzanne S. Gisbertz, Nicole C. T. van Grieken, Eva Berardi, Gianluca Costa

**Affiliations:** 1Department of General Surgery, Sant’ Eugenio Hospital, ASL RM 2, 00144 Rome, Italy; 2Department of Medical Surgical Sciences and Translational Medicine, Sapienza University of Rome, Sant’Andrea Hospital, 00189 Rome, Italy; 3Department of Clinical and Molecular Medicine, Sapienza University of Rome, Sant’Andrea Hospital, 00189 Rome, Italy; 4Department of Surgery, Amsterdam UMC Location University of Amsterdam, 1081 HV Amsterdam, The Netherlands; 5Cancer Center Amsterdam, Cancer Treatment and Quality of Life, 1081 HV Amsterdam, The Netherlands; 6Department of Pathology, Amsterdam UMC Location Vrije Universiteit Amsterdam, 1081 HV Amsterdam, The Netherlands; 7Cancer Biology and Immunology, Cancer Center Amsterdam, 1081 HV Amsterdam, The Netherlands; 8Department of Radiology, San Camillo Hospital, ASL RM 1, 00152 Rome, Italy; 9Department of Life Science, Health and Health Professions, Link Campus University, 00165 Rome, Italy

**Keywords:** radiomics, texture analysis, gastric cancer, oesophageal cancer, oesophagogastric junction cancer

## Abstract

**Simple Summary:**

Oesogastric cancers are often diagnosed at a locally advanced stage, especially in western countries. Thus, their prognosis is highly influenced by correct staging and response rate to preoperative therapy. Radiomics may offer a promising tool for improving the quality of current diagnostics of these tumors. Radiomic predictive models seem to work best when integrated with clinical characteristics. As a future perspective, the incorporation of molecular subgroup analysis to clinical and radiomic characteristics could even increase the effectiveness of these predictive and prognostic models.

**Abstract:**

Background: Oesophageal, gastroesophageal, and gastric malignancies are often diagnosed at locally advanced stage and multimodal therapy is recommended to increase the chances of survival. However, given the significant variation in treatment response, there is a clear imperative to refine patient stratification. The aim of this narrative review was to explore the existing evidence and the potential of radiomics to improve staging and prediction of treatment response of oesogastric cancers. Methods: The references for this review article were identified via MEDLINE (PubMed) and Scopus searches with the terms “radiomics”, “texture analysis”, “oesophageal cancer”, “gastroesophageal junction cancer”, “oesophagogastric junction cancer”, “gastric cancer”, “stomach cancer”, “staging”, and “treatment response” until May 2024. Results: Radiomics proved to be effective in improving disease staging and prediction of treatment response for both oesophageal and gastric cancer with all imaging modalities (TC, MRI, and 18F-FDG PET/CT). The literature data on the application of radiomics to gastroesophageal junction cancer are very scarce. Radiomics models perform better when integrating different imaging modalities compared to a single radiology method and when combining clinical to radiomics features compared to only a radiomics signature. Conclusions: Radiomics shows potential in noninvasive staging and predicting response to preoperative therapy among patients with locally advanced oesogastric cancer. As a future perspective, the incorporation of molecular subgroup analysis to clinical and radiomic features may even increase the effectiveness of these predictive and prognostic models.

## 1. Introduction

Oesophageal, gastroesophageal, and gastric malignancies contribute significantly to cancer-associated morbidity and mortality [1]. For those patients diagnosed with a locally advanced disease, a multimodal therapeutic approach is recommended to increase the chances of survival [2,3,4,5]. In fact, both CROSS and FLOT regimens showed substantial survival benefit when compared to surgery alone or previous chemotherapy regimens [6,7]. However, the impact of therapy on quality of life can be negative and given the significant variation in treatment response, there is a clear imperative to refine patient stratification discriminating between responder and nonresponder patients [8,9,10].

Due to the absence of noninvasive biomarkers, there is a substantial reliance on radiological assessment at various stages of the cancer treatment pathway, including diagnosis, radiotherapy planning, evaluation of treatment response, surveillance, and prognostication. However, the interpretation of radiological images is constrained by human factors, subjective visual interpretation, and can be both time-consuming and prone to inaccuracies. These limitations can be overcome via the introduction of radiomics, which is a tool that can extract quantitative ultrastructural data from medical images, providing features that characterize the spatial relationships of signal intensities within specific tissues and adding a level of visual interpretation that may not be apparent to the human eye [11,12]. Radiomics has indeed proved its utility in a variety of imaging modalities for oesogastric cancer either for disease staging or for predicting response to treatment [13,14,15,16,17,18,19]. Thus, the main goal of radiomics is to overcome the qualitative evaluation of image analysis, usually affected by subjective bias, and to introduce in cancer workflow a quantitative noninvasive imaging biomarker.

Radiomics may also benefit from the integration of machine learning methods to obtain and validate quantitative features, after an accurate feature selection, and extract necessary details to construct predictive models [Figure 1 and Figure 2]. Furthermore, machine learning models can be developed to train, validate, and test datasets for image interpretation, facilitated by the high-throughput extraction of large quantities of data from images [20,21].

This narrative review explores the existing evidence and assesses how radiomics has the potential to improve staging and prediction of treatment response in oesophageal, esophago-gastric, and gastric cancers. Finally, results of personal experience and future perspective are also illustrated.

## 2. Oesophageal Cancer

Traditional radiological staging methods for oesophageal cancer rely on different imaging techniques such as computed tomography (CT), endoscopic ultrasound (EUS), and magnetic resonance imaging (MRI) [22,23,24,25,26]. Fluorine-18 fluorodeoxyglucose positron emission tomography/computed tomography (18F-FDG PET/CT) plays a valuable role in identifying occult distant metastases. Therefore, European guidelines recommend performing 18F-FDG PET/CT in the evaluation of patients considered for esophagectomy. In fact, detecting previously undetected distant metastases can prevent unnecessary surgeries, thus guiding treatment decisions and avoiding futile procedures [2]. These modalities often lack the ability to capture the full heterogeneity of tumors, then the ability of imaging to predict patient prognosis is currently limited. Radiomics has emerged as a promising tool for improving the accuracy of oesophageal cancer staging and to predict tumor regression after preoperative therapy. Several studies have demonstrated the potential of radiomics in characterizing oesophageal tumors and predicting their pathological stages and response to preoperative therapy [13,14,15,16].

### 2.1. CT-Scan

In the radiomics landscape, the most stable imaging method is the CT scan due to its high availability and reduced cost-effectiveness ration, meaning a large amount of exams are commonly available for oesophageal cancer. In fact, several studies built different CT-based radiomic models in oesophageal cancer, as well as Yang et al., who explored the potential of radiomics in predicting the T stage of oesophageal squamous cell carcinoma (ESCC) using CT images. Their study included 116 patients with ESCC and demonstrated that radiomic features extracted from CT images were associated with the T stage of the tumor. The radiomics model showed good predictive performance for both T stage (area under the curve [AUC] of 0.86, sensitivity = 0.77, and specificity = 0.87) and tumor length (AUC = 0.95, sensitivity = 0.92, and specificity = 0.91) [27].

In 2021, Kawahara et al. utilized radiomics combined with machine learning techniques to evaluate tumor differentiation levels using planning CT images from patients diagnosed with locally advanced ESCC. Their study identified thirteen radiomic features capable of distinguishing between poorly differentiated tumors and moderately/well differentiated tumors, achieving an overall accuracy of 85.4%, with specificity and sensitivity rates of 88.6% and 80.0%, respectively [28].

The preoperative detection of lymph node metastases (LNM) plays a crucial role in the management of esophageal cancer patients. In 2017, Wang et al. introduced a support-vector machine (SVM) model designed to detect LNM. Their findings indicated a superior performance of the SVM model compared to conventional CT interpretation, with an area under the curve (AUC) of 0.887 versus 0.705, respectively [29].

Similarly, Tan et al. investigated 1576 radiomic features in a cohort of 230 patients diagnosed with ESCC based on pretreatment CT scans. Their analysis revealed that these radiomic features outperformed size-based image features in predicting LNM, achieving an AUC of 0.758 in the training set and 0.773 in the test set [30].

In a recent meta-analysis, eleven studies were evaluated for the accuracy of CT-based radiomics for prediction of treatment response [16]. Only five studies have been included into the pooled analysis [31,32,33,34,35]. The combined sensitivity and specificity were calculated to be 86.7% and 76.1%, respectively. However, heterogeneity among the studies was found to be significant (I^2^ = 64%) [16].

Most of the results of this analysis originate from Asian institutions, where ESCC is the predominant histological pathology. Therefore, it is essential to exercise caution when extrapolating the relevance of this findings to an international cohort.

### 2.2. MRI

MRI is not commonly included into the preoperative workup of patients with oesophageal cancer. Thus, few data are available on the potential of MRI-based radiomics in the staging of oesophageal malignancies and most of them focus on detecting LNM.

Qu et al. developed and validated an MRI-based radiomics signature for the prediction of LNM in oesophageal cancer patients [36]. The study included a cohort of 181 patients with histologically confirmed oesophageal cancer who underwent preoperative MRI. Radiomic features were extracted from MRI images, nine of them were chosen to form the radiomic signature using machine learning algorithms, which exhibited a significant association with LNM. The AUC for the performance of the radiomic signature was 0.821 in the training cohort and 0.762 in the validation cohort. This signature showed promising results in differentiating between metastatic and nonmetastatic lymph nodes [36].

An interesting step forward in the prediction of LNM in oesophageal cancer patients has been recently achieved by developing a clinical and MRI-based radiomics nomogram [37]. Firstly, the authors showed that relative B7-H3 mRNA expression (obtained via real-time fluorescent quantitative reverse transcription-polymerase chain reaction on preoperative biopsies) was significantly higher in patients with LNM with a good diagnostic performance (AUC: 0.718, sensitivity: 0.733, specificity: 0.706). Secondly, they built an individualized clinical prediction nomogram through multivariable logistic regression analysis including MRI-radiomics features, LN status from CT reports, and B7-H3 mRNA expression. Finally, the nomogram proved both a good diagnostic value (AUC 0.765, sensitivity: 0.800, specificity: 0.706) and a practical value in clinical practice [37].

A recent review draws attention to the quality of these studies, many of which conducted on relatively small sample sizes and were monocentric [38]. Therefore, the applicability and generalizability of these findings necessitate further validation.

Several studies investigate the potential of MRI radiomics signatures to predict the pathological response of ESCC to neoadjuvant chemo or chemoradiotherapy (nCT or nCRT) [39,40,41,42].

In 2018, Hou et al. extracted 138 radiomic features from pretreatment T2W- and SPAIR T2W-MRI of 68 patients with ESCC to build SVM and artificial neural network (ANN) models. The predictive models based on SPAIR T2W sequence showed higher accuracy than those resultant from T2W sequence (SVM: 0.929 vs. 0.893, ANN: 0.883 vs. 0.861) [41].

Lu et al. included 108 ESCC patients in their study, in which radiomics features were extracted from T2-TSE-BLADE sequences and the patients stratified in good responders and poor responders according to the tumor regression grade (TRG). The Delta-model, based on four features out of 107 extracted features, had higher AUC, 0.851 in the training set and 0.831 in the validation set, compared to the pre and postmodels [39].

Recently a multicentric study including 115 patients with ESCC developed and validated a machine learning predictive model based on MR (T2-weighted imaging) radiomics. The model was built on two radiomics features and showed an accuracy of prediction of pathological complete response (pCR) about 82.2%, a sensitivity of 75.0%, and a specificity of 85.7% in the testing set [42].

The role of dynamic contrast-enhanced (DCE)-MRI for the assessment of treatment response in patients with esophageal cancer after neoadjuvant chemoradiotherapy has already been investigated [43]. In 2022, Qu et al. [40] built a DCE-MRI radiomics nomogram to predict the treatment response in esophageal cancer patients. Vascular permeability parameters and 72 radiomics features were extracted from DCE-MRI of 82 patients. Then, three radiomic features were chosen to construct the radiomics signature, demonstrating a significant association with TRG.

The AUC for the combined DCE-MRI radiomics model was 0.84 in the training cohort, while in the validation cohort, it was 0.86. This nomogram exhibited superior discrimination between responders and nonresponders, yielding the highest positive and positive predictive values in both the training and validation sets [40]. To sum up, the application of radiomics to MRI is a promising and future landscape of imaging. Nevertheless, the main limitations are about the reduce populations, retrospective nature of the majority of the studies, and lack of a valid study to test the reliability of radiomics when applied in different scanners with different specifications.

### 2.3. 18F-FDG PET/CT

Even if 18F-FDG PET/CT-based radiomics has already proved its accuracy in predicting clinical and pathological stages of oesophageal cancer before surgery, the majority of studies focused on the prediction of the tumor response to treatment [15,16,44].

Beukinga et al. developed five distinct prediction models for treatment response using a combination of eighteen clinical, geometric, and preprocessed texture features extracted from PET/CT images. Their findings revealed superior predictive performance compared to models exclusively relying on maximum standardized uptake values, highlighting the advantages of utilizing PET/CT radiomic features over conventional parameters [45].

In a pooled analysis of seven studies involving 443 patients by Menon et al., the combined sensitivity and specificity of 18F-FDG PET/CT-based radiomics were determined to be 86.5% and 87.1%, respectively. Notably, there was substantial evidence of heterogeneity between the studies (I^2^ = 72%) [16]. In their study, the evaluation of treatment response across different imaging modalities revealed that 18F-FDG PET/CT imaging exhibited a higher specificity compared to CT scans (87.1% vs. 76.1%). However, sensitivities were comparable between the two modalities (86.5% vs. 86.7%).

Earlier research has revealed that the integration of clinical parameters with radiomic features derived from 18F-FDG PET/CT enhances the predictive capacity for predicting pCR [46].

Besides clinical and metabolic parameters, integrating radiomic features with biological expression products can enhance the accuracy of radiomic models [37]. Incorporating them into a comprehensive prediction model can improve its performance. Beukinga et al. demonstrated this by including human epidermal growth factor receptor 2 (HER2) and CD44 in a 18F-FDG PET-based clinic-radiomic model, resulting in improved overall performance in predicting nCRT response in EC patients (AUC, 0.857) [47].

Combining different imaging modalities (CT and PET scans) may be optimal in discerning the response to treatment, as shown by Rishi et al. with an AUC of 0.87 [34].

## 3. Gastroesophageal Junctional Cancer

In contrast to the multitude of articles on radiomics in oesophageal and stomach cancer, the literature data on gastroesophageal junction (GEJ) cancer are very limited. In addition, data extraction from large cohort studies on radiomics in gastric or oesophageal cancer that include cardiac localization may be inaccurate.

A recent systematic review by Mori et al. focusing on the role of radiomics in staging and restaging GEJ adenocarcinomas was able to include only two studies [48].

Wang et al. examined the potential of pretreatment CT-based radiomic features to determine overall prognosis and predict pCR in 146 patients with stage II/III GEJ adenocarcinoma undergoing nCRT [49]. Among the features analyzed, only shape compactness was found to be significantly associated with overall survival. The resulting pretreatment risk scoring system, which combined shape compactness with pathological grade differentiation, showed a significant negative association with pCR [49].

Chang et al., in their monocentric study including 200 patients with GEJ adenocarcinoma, explored the effectiveness of CT-based radiomic features in distinguishing between T3 and T4a during clinical staging [50]. They developed a radiomic score (Radscore) incorporating 11 radiomic features, which demonstrated strong discriminative capability (AUC: 0.812, sensitivity: 0.915, specificity: 0.538). This discriminative power was further enhanced when combined with conventional CT scan features in an integrated nomogram (AUC: 0.812, sensitivity: 0.936, specificity: 0.692) [50].

However, the relatively low radiomics quality score values of both studies (Wang:15/31 and Chang: 19/31) highlighted the limited clinical applicability of these models, primarily due to their retrospective nature and insufficient validation [48].

Recently, a study by Du et al. explored the application of CT-based radiomics for distinguishing between adenocarcinoma and squamous cell carcinoma at GEJ [51]. The authors developed and validated six radiomic models that demonstrated promising accuracy in the differential diagnosis of these two cancer types. The 3D-arterial-venous combined reached the best accuracy (0.841 in the training group [TG] and 0.808 in the validation group [VG]), AUC (TG: 0.904, VG: 0.901), sensitivity (TG:0.802, VG: 0.795) and specificity (TG: 0.879, VG: 0.821) [51]. Thus, the role of radiomics in this entity is absolutely promising, but further investigations with multicenter studies is essential to make the radiomic approach stable and reproducible.

## 4. Gastric Cancer

Traditional radiological staging of gastric cancer mainly relies on CT-scan. EUS can play a role in the assessment of T and N stage in potentially operable tumors. The 18F-FDG PET/CT method can be useful for detecting metastases; however, its use is not routinely recommended. MRI is not even mentioned in the last edition of the European Society for Medical Oncology (ESMO) [3].

Radiomics is emerging as a promising tool for implementing characterization and stratification of gastric cancers. Most studies focus on the potential of radiomics to predict tumor and lymph nodes response to preoperative chemotherapy [17,18,19].

### 4.1. CT-Scan

As well as in oesophageal cancer, also in gastric cancer, CT has the key role in staging and restaging before the surgical approach. The high availability of examinations and the ability to homogenize medical images make CT the main imaging method for testing radiomics in gastric cancer.

The first potential of radiomics to be investigated in gastric cancer was the texture-based classification. Two studies have shown that examining both first- and second-order radiomics features in contrast-enhanced CT images can assist in differentiating lymphoma from gastrointestinal stromal cancer or adenocarcinoma [52,53]. Focusing on adenocarcinoma, Liu et al. demonstrated, in a cohort of 107 patients, the potential of CT-based texture analysis in predicting Lauren classification, tumor differentiation, and vascular invasion [54].

Lymphovascular invasion (LVI) has also been investigated among the possible prediction fields of radiomics [55]. The authors integrated clinical and radiomics feature through deep transfer learning into a model for predicting LVI. This model showed good predictive performance in both training (AUC: 0.755) and testing dataset (AUC: 0.725) [55].

Lymph node status is universally recognized as an independent prognostic factor of gastric cancer prognosis [56,57]. For this reason, it is essential to determine the presence of positive lymph nodes at diagnosis, so as to determine the best course of treatment for the patient.

Radiomics proved to be useful for this application in a large cohort study in which the authors developed a CT-based deep learning radiomic nomogram for the preoperative assessment of lymph node metastases [17]. This nomogram was built combining radiomics signatures with clinical characteristics of 730 patients with locally advanced gastric cancer (LAGC) from five centers in China and one center in Italy. It proved to be effective in preoperatively identifying lymph node metastasis and it outperformed the commonly used clinical N stages, tumor size, and clinical model. Additionally, it showed a significant correlation with the overall survival [17].

Another computational clinical decision support system (DSS) was developed for the same purpose using machine learning-based analysis [18]. The AUC for predicting LNM was 0.824 and 0.764 in the training and validation set, respectively. The DSS outperformed the conventional staging criteria in predicting LNM, with a higher accuracy in both training and testing datasets [18].

Recently, radiomics was tested as quantitative tool to predict MSI status in gastric cancer and the group of Chen et al. demonstrated that radiomics features, extracted from preoperative contrast enhanced CT scans, combined with clinical and semantic radiological features could be able to predict MSI status with an AUC of 0.83. Their analysis included both intratumoral and peritumoral radiomic features, showing the importance of peritumoral fat in gastric cancer [58].

Regarding the prediction of treatment response, a recent metanalysis included 3373 patients with LAGC treated with preoperative chemotherapy [19]. The main finding of the pooled analysis was an improved predictive performance of radiomics features when integrated with clinical data (c-index: 0.760 [radiomics features], 0.610 [clinical features], and 0.802 [integrated model]) [19]. Interestingly, among the fourteen studies included in the metanalysis, all came from China except one that came from Italy. These results should be interpreted with caution because they may not be directly applicable to western patients.

### 4.2. MRI

Even if MRI is not commonly used in the diagnostic pathway of gastric cancer staging, there are several studies showing its potential in risk stratification and prediction of pathological response to preoperative chemotherapy.

Chen et al. extracted 1305 radiomics features from preoperative 3.0 T MRI of 146 patients with LAGC and they found that a two-radiomic signature was significantly associated with LNM [59]. The authors constructed a DWI-based radiomic nomogram integrating the radiomic signature, the clinical minimum apparent diffusion coefficient, and MRI-reported N staging for prediction of LNM. The nomogram showed good predictive performance both in the training (AUC: 0.850) and in the external validation cohort (0.878) [59].

MRI proved to be effective in predicting pathological response to preoperative chemotherapy in both single sequence-based Radscores and multiparametric (mp)-MRI radiomics nomogram [60].

However, the MRI-based radiomics nomogram reached the best predictive ability with an AUC of 0.844 in the training group and 0.820 in the validation group [60].

The same authors further explored the potential of MRI-based radiomics for prediction of tumor response to treatment comparing and combining it to CT-based radiomics models [61].

The comparison between mp-MRI and CT model showed similar predictive performances between the two imaging methods in the training (0.831 vs. 0.770, *p* = 0.267) and validation dataset (0.797 vs. 0.746, *p* = 0.137). The multimodal nomogram combining CT and mp-MRI Radscores achieved the highest AUC in both training (0.893) and validation (0.871) cohorts [61].

Tumor-infiltrating lymphocytes (TILs) within the tumor microenvironment has been proven to have prognostic value in gastric cancer patients [62]. Indeed, high levels of CD3+ TILs were significantly associated with improved survival. Furthermore, CD3+ TILs were found to correlate with EBV-positive and PD-L1-positive gastric cancer, potentially aiding in the identification of targets for immunotherapy [62].

Huang et al. have first demonstrated that radiomic features from DCE-MRI are linked to the expression of TILs CD8+ and CD4+ in LAGC [63]. Then they showed how DCE-MRI-based machine learning classifiers can accurately predict the expression levels of CD3+, CD4+, and CD8+ tumor-infiltrating lymphocytes in patients with LAGC [64].

Since high levels of CD3+ TILs are associated with improved survival, MRI-based radiomics could serve as an alternative and noninvasive prognostic biomarker in GC.

### 4.3. 18F-FDG PET/CT

The main application of 18F-FDG PET/CT in the staging of gastric cancer patients is the detection of occult peritoneal or nodal metastases [3,65,66,67,68]. However, several studies investigated its potential in discriminating the histology of the tumor, the expression of human epidermal growth factor receptor 2 (HER2), the presence of LVI, and the expression of tumor immune microenvironment [69,70,71,72,73,74,75].

The detection of peritoneal or distant metastasis is the primary indication for performing a 18F-FDG PET/CT on a gastric cancer patient. However, the literature presents conflicting results between eastern and western studies highlighting differences in both patients and tumor characteristics.

A Chinese group developed a nomogram based on 18F-FDG PET/CT including fourteen radiomics features and clinical risk factors (peritoneal effusion, mean standardized uptake value, and carbohydrate antigen 125). Their combined nomogram performed very well in detecting peritoneal metastasis with an AUC of 0.92 in both training and validation cohorts [76].

On the other hand, the Dutch prospective multicenter PLASTIC study concluded that 18F-FDG PET/CT-based radiomics did not aid in the preoperative identification of peritoneal and distant metastases in patients with LAGC. In fact, none of their models (clinical, radiomic, and clinico-radiomic models) showed good predictive performance (AUCs of 0.67, 0.60 and 0.71, respectively) [77].

Nodal staging remains crucial in the diagnostic framework of gastric cancer. Several 18F-FDG PET/CT-based radiomics nomograms have been generated [65,66,67,68]. All of them showed promising results in terms of LNM detection with AUCs ranging from 0.751 to 0.897 and decision curve analysis showing clinical usefulness.

Interestingly, Xue et al. focused their analysis on the prediction of N2-3b lymph node metastasis. Their combined radiomics nomogram, comprising 18F-FDG PET/CT radiomics features and carcinoembryonic antigen level, can be effectively utilized for personalized prediction of high-risk N2-3b metastasis (AUC: 0.81) [68].

Among primary gastric lymphomas, diffuse large B-cell lymphomas tend to be more aggressive and are commonly identified at a later stage with generally a worse prognosis. The technique of 18F-FDG PET/CT radiomics has been shown to be useful in discriminating diffuse large B-cell lymphomas from mucosa-associated lymphoid tissue (MALT) with high accuracy [69].

HER2 expression in LAGC is essential for guiding clinical decisions, target therapy, and advancing the field of precision medicine. Several studies have demonstrated the predictive power of the radiomic model based on 18F-FDG PET/CT for the detection of HER2 expression, with AUCs ranging from 0.628 to 0.993 [70,71,72].

LVI is linked to metastasis and poor survival outcomes in gastric cancer patients, but noninvasive diagnosis of LVI remains challenging. Two studies constructed predictive models incorporating CT and PET/CT images alongside clinical variables, to preoperatively predict the LVI status in gastric cancer [73,74]. In both studies the combined models demonstrated strong calibration and clinical utility for predicting LVI. Focusing on the imaging models, PET/CT integrated models were superior to PET- and CT-based models [73,74].

The tumor immune microenvironment of gastric cancer may offer valuable prognostic and predictive information, such as clinical outcomes and response to adjuvant treatment. Li et al. extracted 80 features from 18F-FDG PET/CTs of 230 patients with gastric cancer. They built and validated an 8-feature radiomics model to predict the tumor immune microenvironment (AUC. 0.692 [testing set] and 0.713 [validating set]). Moreover, a significant association between high radiomics tumor immune microenvironment score and response to adjuvant chemotherapy was found [75].

## 5. Results of Personal Experience

This research is part of an international postgraduate program among Italian and Dutch Universities and facilities. We conducted a retrospective analysis of a prospective maintained database. Between 2019 and 2020 all patients with resectable LAGC who underwent perioperative chemotherapy using the FLOT regimen (5-fluorouracil, leucovorin, oxaliplatin, and docetaxel) and surgical resection at the Gastrointestinal Surgery Unit of Sant’Andrea University Hospital were included in a preliminary study [78]. The TRG was evaluated according to Becker classification: TRG 1a-1b were considered responders and TRG 2-3 nonresponder patients [79]. For each patient, volumetric tumor segmentation was manually performed on both the baseline and postchemotherapy (PostChT) CT scans in consensus by two radiologists utilizing 3D slicer version 4.10.2 software.

Among the 48 patients with LAGC treated by preoperative chemotherapy, only fifteen patients fulfilled the inclusion criteria: six responders and nine nonresponders.

Radiomics features from prechemotherapy CT scans revealed significant differences between responders and nonresponders, particularly in Shape (LeastAxisLength), GLCM (Cluster Shade and Autocorrelation), First Order (Skewness), and NGTDM (Strength) features. The diagnostic performance of these features was evaluated using ROC curves, which demonstrated AUCs of 0.815 for LeastAxisLength, 0.907 for both Cluster Shade and Autocorrelation, 0.889 for Skewness, and 0.815 for Strength (all *p* < 0.011) (Table 1).

ΔRadiomics (pre and postChT) values demonstrated significant differences between responders and nonresponders in Shape (MeshVolume, LeastAxisLength, SurfaceVolume), GLRLM (LongRunEmphasis), GLSZM (LargeAreaLowGrayLevelEmphasis), and NGTDM (Contrast) features. Additionally, ROC curve analysis indicated significant results for these radiomic features, with an AUC of 0.889 for MeshVolume, 0.833 for LeastAxisLength, 0.852 for SurfaceVolume, 0.889 for LongRunEmphasis, 0.833 for LargeAreaLowGrayLevelEmphasis, and 0.796 for Contrast (all *p* < 0.007) (Table 1).

This preliminary analysis highlighted the promising role of radiomics in predicting TRG in gastric adenocarcinoma patients treated with perioperative FLOT chemotherapy. This study formed the rationale for a subsequent multicenter research study on a radiomics-molecular predictive model or GEJ and gastric cancer TRG (NCT04878783) that is still ongoing.

## 6. Limitations

It is necessary to point out that radiomics is a relatively new field of study that still needs validation before being incorporated into clinical practice. Despite the interesting findings of the studies analyzed in this narrative review, radiomics in oesogastric cancers still has several limitations.

First, the retrospective nature of the studies and the relatively small simple size could lead to a possible instability of the results. Secondly, most of the studies are monocentric and come from eastern countries. This leads to two orders of problems: the lack of reproducibility of data and the difficulty of applying the results obtained to a western population. Third, multicentric studies lead to difficult data harmonization because of inter-patient (e.g., eastern v. western population) or inter-scanner variability due to different voxel size and different acquisition protocols. Data harmonization is a major limitation of radiomics, several methods have been proposed to overcome the high variability of quantitative features with the aim to make the studies reproducible and robust. The most recent approach that is gaining a consistent role is combatting batch effect, which can reduce differences between voxel sizes more consistently, showing promising results in CT, MRI, and PET/CT [80,81]. However, it is a matter a scientific debate whether homogeneity of radiomics data is an advantage or a drawback into multicentric setting.

Finally, it is worth mentioning that radiomics cannot replace but possibly complement histopathological evaluation, which remains the gold standard for cancer diagnosis.

## 7. Conclusions

Radiomics shows potential in noninvasive staging and predicting response to preoperative therapy among patients with locally advanced oesogastric cancer. Incorporating clinical and radiomic features into models improves both diagnostic capabilities and the ability to predict tumor regression. However, there is still a long way to go to incorporate radiomics into a clinical decision-making process. It is essential to overcome the main radiomics limitations due to low reproducibility of data linked to the inter-scanner variability, and the lack of consistent prospective studies, which could validate radiomic results. Then, further large-scale multicenter research comparing quantitative models and the creation of imaging-derived tools for effective risk stratification are essential to establish a solid and reliable evidence base.

## 8. Future Directions

As a future perspective, the incorporation of molecular subgroup analysis to clinical and radiomic features may even increase the effectiveness of these predictive and prognostic models.

Moreover, prospective and multicentric studies, including patients from both eastern and western countries are needed to overcome the actual limitations of the application of radiomics into clinical practice.

## Figures and Tables

**Figure 1 cancers-16-02664-f001:**
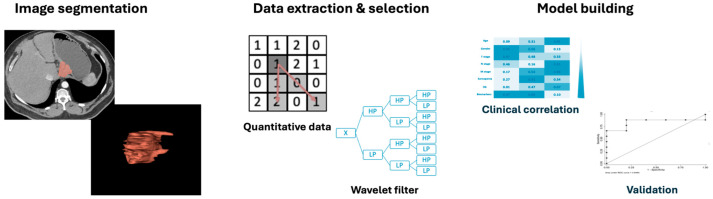
Esophago-gastric junction cancer radiomics workflow. The red arrow is a general graphical representation of the correlation of radiomics features with each other based on pixel intensity.

**Figure 2 cancers-16-02664-f002:**
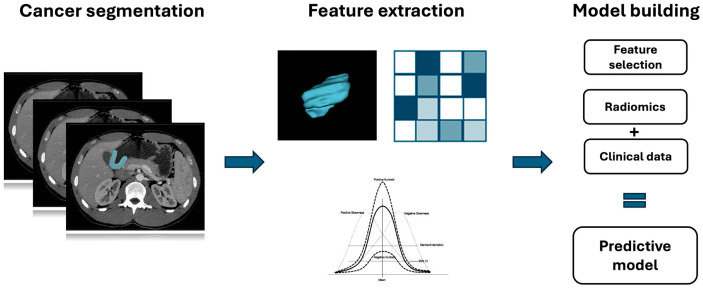
Gastric cancer radiomics workflow.

**Table 1 cancers-16-02664-t001:** PreCHT and ΔRadiomics analysis.

PreCHT Radiomics Features		ROC Curve Analysis				
		AUC	Sensibility	Specificity	95%C.I.	*p*
Shape	LeastAxisLength	0.815	88.89%	66.67%	0.53–0.96	0.011
GLCM	Cluster Shade	0.907	66.67%	100%	0.64–0.99	<0.0001
	Autocorrelation	0.907	88.89%	83.33%	0.64–0.99	<0.0001
First order	Skewness	0.889	88.89%	83.33%	0.62–0.99	<0.0001
NGTDM	Strength	0.815	55.56%	100%	0.53–0.96	0.007
ΔRadiomics Features						
Shape	Mesh Volume	0.889	66.67%	100%	0.62–0.99	<0.0001
	LeastAxisLength	0.833	77.78%	100%	0.55–0.97	0.0045
	SurfaceVolume	0.852	88.89%	83.33%	0.57–0.97	0.0021
GLRLM	LongRunEmphasis	0.889	66.7%	100%	0.62–0.99	<0.0001
GLSZM	LargeAreaLowGrayLevelEmphasis	0.833	100%	66.67%	0.55–0.97	0.007
NGTDM	Contrast	0.796	66.67%	83.33%	0.53–0.96	0.005

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
