# Peer review of "Radiomics in Oesogastric Cancer: Staging and Prediction of Preoperative Treatment Response: A Narrative Review and the Results of Personal Experience"

_cancers, 2024, doi:10.3390/cancers16152664_

Round 1

Reviewer 1 Report

Comments and Suggestions for Authors

In this study, the authors investigated the potential of radiomixin for the staging of esophageal, GEJ and gastric cancers and the prediction of preoperative treatment response. The study reported that radiomixin is effective in staging these cancers and predicting treatment response. Here, it is suggested that radiomix models perform better when different imaging modalities are integrated and clinical features are combined with radiomix features. 

The main motivation of this study is that Radiomix can optimize treatment planning by predicting patients' responses to treatment and help avoid unnecessary surgical procedures. In this sense, this study can make an important contribution to the literature. 

The results of the study suggest that radiomix has the potential to make a significant contribution to the staging of esophageal and gastric cancers and the prediction of treatment response.

I believe that the use of radiomixin in CT, MRI, 18F-FDG PET/CT models will improve diagnostic capabilities and the capacity to predict tumor regression, in this sense, the study is important. However

1- The generalization of the present findings to international cohorts needs clarification (because most studies were conducted by institutions in Asia).

2- In particular, the retrospective nature of the study and small sample sizes limit the generalizability of the findings, which should be explained in detail.

3- The safety of radiomixin should be tested in prospective studies.

4- This data harmonization should be explained in detail in order to ensure data harmonization between centers with different scanners and technical features. 

5- Although the authors have made a short review, a discussion section should be added to the study, comparing the studies in the literature and indicating the importance of this study. 

6- Personal experiences should be presented in a table to make it easier for readers to understand. 

7- The sample hospital data given in the personal experinces section should be presented in a table and the values given should be explained by comparing the metric values. 

Reviewer 2 Report

Comments and Suggestions for Authors

The article titled "Radiomics in Oesogastric Cancer: Staging and Prediction of Preoperative Treatment Response. A Narrative Review and Results of Personal Experience" by Giovanni Maria Garbarino et al. offers valuable insights. However, there are several areas that require attention:

1. The present review describes that most of the existing studies on radiomics come from Asian institutions, which raises questions about the generalizability of the findings to other populations worldwide.

2. Although the authors highlight the potential of radiomics to improve tumour characterisation and predict treatment response, they did not address the potential limitations of radiomics as a relatively new field of study.

3. It is essential to note that radiomics is not a replacement for histopathological evaluation, which remains the gold standard for cancer diagnosis and staging.

4. It would be beneficial for the authors to provide a comprehensive table summarising radiomics studies for oesophageal and gastric cancers, predicting their pathological stages and treatment.

5. The authors could briefly discuss incorporating molecular subgroup analysis into clinical and radiomics features to improve predictive and prognostic models.

6. Despite being a review, the author must write the aim, objectives, and conclusion of the study in the abstract. Authors must write the abstract according to the author's instructions.

Comments on the Quality of English Language

Extensive editing of English language required

Round 2

Reviewer 2 Report

Comments and Suggestions for Authors

Accept in present form

Comments on the Quality of English Language

Minor editing of English language required